# Exploring the Toxicity and Therapeutic Potential of *A. dahurica* and *A. pubescens* in Zebrafish Larvae: Insights into Anxiety Treatment Mechanisms

**DOI:** 10.3390/ijms26072884

**Published:** 2025-03-22

**Authors:** Mariola Herbet, Jarosław Widelski, Marta Ostrowska-Leśko, Anna Serefko, Krzysztof Wojtanowski, Joanna Kurek, Iwona Piątkowska-Chmiel

**Affiliations:** 1Chair and Department of Toxicology, Faculty of Pharmacy, Medical University of Lublin, Jaczewskiego 8b Street, 20-090 Lublin, Poland; marta.ostrowska-lesko@umlub.pl (M.O.-L.); joanna.kurek07@gmail.com (J.K.); iwona.piatkowska-chmiel@umlub.pl (I.P.-C.); 2Department of Pharmacognosy with Medicinal Plants Garden, Medical University of Lublin, 1 Chodźki Street, 20-093 Lublin, Poland; jwidelski@pharmacognosy.org (J.W.); krzysztof.wojtanowski@umlub.pl (K.W.); 3Department of Clinical Pharmacy and Pharmaceutical Care, Medical University of Lublin, 1 Chodźki Street, 20-093 Lublin, Poland; anna.serefko@umlub.pl

**Keywords:** *Angelica dahurica*, *Angelica pubescens*, zebrafish larvae, toxicity, gene expression, anxiolytic effects

## Abstract

This study assessed the toxicity and therapeutic potential of *Angelica dahurica* and *Angelica pubescens* using *Danio rerio* (zebrafish) larvae. Toxicity was evaluated through mortality, malformations, and gene expression changes related to stress and the HPA axis. *A. dahurica* demonstrated low toxicity (LD50 (50% lethal dose) >200 µg/mL), with no significant malformations at 15–30 µg/mL, although higher doses caused edemas and heart defects. *A. pubescens* exhibited higher toxicity, with 100% mortality at 200 µg/mL and severe malformations. Both species showed potential cardiotoxicity, slowing heart rates after prolonged exposure. Gene expression studies suggested *A. dahurica* had stress-protective effects, increasing *nr3c1* expression, while *A. pubescens* had dose-dependent effects, with lower concentrations having anxiolytic properties and higher concentrations increasing stress. Interestingly, diazepam showed unexpected gene expression changes, highlighting the influence of environmental and dosage factors. In conclusion, both species show therapeutic potential for anxiety, with *A. dahurica* showing promising effects at lower concentrations. However, *A. pubescens* requires careful dosage management due to its higher toxicity risks. Further studies are needed to optimize therapeutic applications and fully understand mechanisms of action.

## 1. Introduction

Anxiety disorders are currently the most prevalent mental health conditions worldwide, with onset typically occurring during early adulthood or adolescence [1]. It is estimated that approximately one-third of the population will experience these disorders in their lifetime, with a prevalence of around 7.3% (ranging from 4.8% to 10.9%). The most common anxiety disorders include specific phobias and panic disorder (with or without agoraphobia), followed by social phobia and generalized anxiety disorder [2]. These disorders are associated with dysfunctions in brain circuits that respond to perceived threats. The factors predisposing individuals to anxiety disorders are both genetic and environmental, with significant epigenetic interactions involved in their pathogenesis [3]. Anxiety disorders are a major cause of functional impairment, typically following a chronic and persistent course, and their treatment often proves inadequate. Globally, anxiety disorders remain frequently undiagnosed and untreated. Treatment strategies for anxiety disorders include evidence-based psychotherapy and pharmacotherapy. Despite the introduction of several new substances into clinical trials, no new pharmacological agents for the treatment of anxiety have been developed in over 20 years. Furthermore, despite promising results in rodent studies, the efficacy of these substances in human clinical trials remains unproven [4].

Pharmacotherapy for anxiety disorders primarily targets the GABAergic, serotonergic (5-HT), glutamatergic, neuropeptide, and endocannabinoid systems [4]. Benzodiazepines (BDZs) (e.g., chlordiazepoxide, diazepam) are classical GABAergic agents that have served as a model for other anxiolytic drugs. BDZs bind to GABAA receptors, inhibiting neurotransmission in the brain, and are widely used to treat various anxiety disorders. One of their undeniable advantages is their rapid onset of action, which is particularly important in the initial stages of treatment before other medications are introduced. However, BDZs have significant side effects, including tolerance, dependence, sedation, and memory impairment [5]. Pharmacotherapy for mild to moderate social anxiety disorder, panic disorder, agoraphobia, and generalized anxiety disorder typically involves selective serotonin reuptake inhibitors (SSRIs) and serotonin-norepinephrine reuptake inhibitors (SNRIs) [6]. Nevertheless, treatment responses are often suboptimal, and the currently used drugs are associated with numerous adverse effects.

In recent years, especially in connection with the COVID-19 pandemic, there has been a significant rise in depression rates. A relationship has also been observed between the co-occurrence of anxiety disorders, especially generalized anxiety disorder, and depression [7]. Anxiety disorders often precede the onset of depressive disorders [8]. A global study found that 45.7% of individuals with major depressive disorder had one or more anxiety disorders during their lifetime [9]. There are hypotheses suggesting shared pathophysiological mechanisms, and common structural and functional brain alterations in various mental disorders, including anxiety and major depression. Evidence indicates that chronic stress, oxidative stress, and neuroinflammatory processes play a key role in the pathology of these conditions [10]. The co-occurrence of anxiety and depression complicates the treatment of these comorbid conditions, making it even more challenging. Therefore, there is a pressing need to explore novel therapeutic strategies. Natural substances, owing to their multifaceted properties and high safety profile, present an intriguing target for such strategies.

*Angelica pubescens*, known as Duhuo in Traditional Chinese Medicine (TCM), contains numerous compounds with antioxidant, analgesic, antipyretic, antibacterial, antifungal, and anti-inflammatory effects [11]. Ostenol, a coumarin derived from the root of *Angelica pubescens*, inhibits monoamine oxidase A (hMAO-A), suggesting its potential in treating depression [12]. Another coumarin, osthole, exhibits a broad range of activities and could serve as a potential treatment for various diseases, including osteoporosis and fatty liver disease [13]. This compound has also been shown to possess anti-inflammatory and neuroprotective effects, protecting against neuronal death, cognitive deficits, and brain swelling. Moreover, osthole inhibits microglial proliferation in epilepsy models, which are characterized by excessive activation of the PI3K/Akt/mTOR pathway [14]. *Angelica dahurica*, known as Baizhi in TCM [15], also contains a variety of active coumarins, such as imperatorin (IMP), isoimperatorin, and auraptenol (AP), which exhibit antibacterial, anticancer, anti-inflammatory, and analgesic properties [16]. Studies have shown that AP exhibits potential therapeutic effects in mental illnesses like depression, with its antidepressant effects comparable to those of imipramine in validated mouse models. Additionally, IMP has been shown to impact the serotonergic system and protect against depressive-like behaviors in rodent models [17].

The aim of the present study was to assess the effects of plant extracts from *A. dahurica* and *A. pubescens* on anxiety behavior in the *Danio rerio* model. *Danio rerio* shares over 70% genetic similarity with humans, making it an appropriate model for studying the mechanisms underlying anxiety and depression [18]. In this study, the following experiments were conducted: (1) toxicity studies to evaluate the safety profile of *A. dahurica* and *A. pubescens* extracts, (2) behavioral studies to assess the anxiolytic properties of these extracts, and (3) molecular studies to evaluate the effects of *A. dahurica* and *A. pubescens* extracts on the expression of genes involved in the mechanisms of anxiety and/or depression. Orexins (HCRT-1 and HCRT-2) are neuropeptides that play a crucial role in regulating wakefulness, sleep, stress behavior, appetite, and reward [19]. Studies in zebrafish (*Danio rerio*) have shown that increased *hcrt* gene expression leads to prolonged wakefulness and fragmented sleep, underscoring this gene’s significant impact on activity cycles [20]. Similarly, CRHR1, the receptor for corticotropin-releasing hormone, is a key component of the hypothalamic–pituitary–adrenal (HPA) axis and plays a vital role in managing anxiety and stress responses [21]. CRHR1 deficiency is linked to reduced anxiety-like behavior, whereas its overexpression correlates with chronic stress.

The expression of the *nr3c1* and *nr3c2* genes, which encode glucocorticoid and mineralocorticoid receptors, respectively, also plays a critical role in regulating stress and anxiety-related behaviors [22]. Dysregulation of these genes has been associated with various pathological conditions, such as depression and anxiety disorders. In the context of research on potential anxiolytic therapies, other important genes include *ccka*, which is associated with dopamine regulation and anxiety-like behaviors [23], and *avp*, whose expression correlates with the intensity of stress and anxiety responses [24].

The search for natural substances as an alternative or support in neuropsychiatric treatment is of growing interest, especially in anxiety and stress-related disorders. *A. dahurica* and *A. pubescens*, widely used in traditional medicine, are known for their putative therapeutic properties, but the mechanisms underlying their action and their safety profiles remain insufficiently understood. The present study addresses these gaps by using zebrafish larvae as a new in vivo model, enabling a comprehensive assessment of both morphological toxicity and molecular responses, including changes in gene expression related to the hypothalamic–pituitary–adrenal (HPA) axis. By assessing dose-dependent effects on development, cardiotoxicity, and stress modulation, this study provides new insights into the dual potential of these compounds, highlighting their role in safe therapeutic use. These findings contribute to a deeper understanding of plant-based treatments, offering a basis for optimizing their use in treating neuropsychiatric conditions.

## 2. Results

### 2.1. Chemical Profile of Methanolic Extracts from A. dahurica and Angelica pubescens

Qualitative analysis of methanolic extracts from roots of *A. dahurica* and *A. pubescens* revealed that the studied plants are rich sources of furanocoumarins and simple coumarins, which occur together with furanocoumarin compounds, respectively. In total, nine compounds were identified from the methanolic extract of *A. dahurica* by high-performance liquid chromatography–electrospray ionization–quadrupole time-of-flight–mass spectrometry (HPLC/ESI-QTOF-MS), such as oxypeucedanin hydrate, psoralene, 8-methoxypsolarene (xanthotoxin), oxypeucedanin, byakangelicol, heraclenin, imperatorin, phellopterin, and isoimperatorin, all representing furanocoumarins (Figure 1, Table 1).

In the case of the methanolic extract of *A. pubescens* root (Figure 2, Table 2), high-performance liquid chromatography–electrospray ionization–quadrupole time-of-flight–mass spectrometry (HPLC/ESI-QTOF-MS) allowed for the identification of 10 compounds representing simple coumarins (auraptenol, angelol G, 7-methoxy-5-prenyloxycoumarin, and osthol), furanocoumarins (nodakenetin, imperatorin, phellopterin, and isoimperatorin), and dihydrofuranocoumarins (columbianetin acetate and angenomalin).

### 2.2. Toxicity Studies

The results of toxicity studies are summarized in Table 3. An exposition to the highest concentrations of *A. pubescens* extract (i.e., 200 and 100 µg/mL) resulted in increased mortality of zebrafish embryos/larvae as compared to the control groups (incubated in E3 medium and 1% solution of DMSO). All subjects exposed were dead after a 24 h incubation period in a solution of 200 µg/mL, whereas a 5-day exposure to a solution of 100 µg/mL caused the death of 75% of embryos/larvae. The mortality of zebrafish embryos/larvae incubated in lower concentrations of *A. pubescens* extract (i.e., 50, 30, and 15 µg/mL) and in all tested dilutions of *A. dahurica* extract (i.e., 200, 100, 50, 30, and 15 µg/mL) was comparable to the one detected in the control groups.

Malformations/deformities of the head, eyes, tail, heart, spine, yolk, and/or reduction in pigmentation were observed in all tested groups. However, a significantly higher number of subjects incubated in solutions of *A. dahurica* (200, 100, 50, and 30 µg/mL) and *A. pubescens* (100, 50, and 30 µg/mL) extracts had developmental abnormalities when compared to the control groups. Only the lowest tested dilutions of both extracts (i.e., 15 µg/mL) were as safe as E3 medium and 1% solution of DMSO to zebrafish embryos/larvae.

After 96 h of exposure to *A. dahurica* extract (200, 100, 50, 30, and 15 µg/mL), zebrafish larvae presented a significantly lower activity of the heart when compared to control groups exposed to E3 medium or 1% solution of DMSO. Statistical analysis confirmed significant differences between groups: F (6, 142) = 33.24; *p* < 0.0001 (Figure 3A). Similarly, the heart rate of zebrafish after a 96 h incubation in solutions of *A. pubescens* extract (100, 50, 30, and 15 µg/mL) was significantly lower in measurement than values obtained for the control groups. One-way analysis of variance confirmed the differences between the studied groups: F (5, 112) = 8.968; *p* < 0.0001 (Figure 3B). The high mortality of embryos/larvae in the 200 µg/mL solution made it impossible to measure heart rate in this group of *Danio rerio*.

### 2.3. Molecular Studies

In the group of *Danio rerio* treated with diazepam, a statistically significant increase in *hcrt* gene expression (Figure 4A) was observed compared to the DMSO control group (*p* < 0.001). Similarly, significant increases in *hcrt* gene expression were detected in the following treatment groups: *A. pubescens* 1.5 µg/mL, 6 µg/mL, and 9 µg/mL (*p* < 0.001 for all). In contrast, a significant decrease in *hcrt* gene expression was observed in the *A. dahurica* 9 µg/mL group (*p* < 0.01). When compared to the diazepam-treated group, *hcrt* gene expression was significantly lower in all treatment groups: *A. dahurica* 1.5 µg/mL, 6 µg/mL, and 9 µg/mL, and *A. pubescens* 1.5 µg/mL, 6 µg/mL, and 9 µg/mL (*p* < 0.001 for all).

Regarding *crhr1* gene expression (Figure 4B), significant reductions were observed in the *A. dahurica* 1.5 µg/mL, 6 µg/mL, and 9 µg/mL groups (*p* < 0.001, *p* < 0.001, *p* < 0.001, respectively) compared to the DMSO control. A slight reduction was also noted in the diazepam-treated group and in the *A. pubescens* 6 µg/mL and 9 µg/mL groups (*p* < 0.01, *p* < 0.001, *p* < 0.001, *p* < 0.05, respectively). Compared to the diazepam-treated group, the *A. dahurica* 1.5 µg/mL, 6 µg/mL, and 9 µg/mL groups exhibited significant decreases in *crhr1* expression (*p* < 0.001, *p* < 0.01, *p* < 0.001, respectively), with a slight change in the *A. pubescens* 1.5 µg/mL group (*p* < 0.05).

For *ccka* gene expression (Figure 4C), a decrease was noted in the diazepam and *A. dahurica* 1.5 µg/mL groups (*p* < 0.001, *p* < 0.001, respectively) compared to the DMSO control. Conversely, significant increases were found in the *A. pubescens* 6 µg/mL and 9 µg/mL groups (*p* < 0.001 for both) compared to the DMSO control. When compared to the diazepam-treated group, increased *ccka* gene expression was observed in the *A. dahurica* 6 µg/mL, 9 µg/mL, *A. pubescens* 1.5 µg/mL, 6 µg/mL, and 9 µg/mL groups (*p* < 0.01, *p* < 0.001, *p* < 0.001, *p* < 0.001, *p* < 0.001, respectively). Regarding *ist1* gene expression (Figure 4D), a significant increase was observed in the diazepam and *A. pubescens* 9 µg/mL groups (*p* < 0.001 and *p* < 0.01, respectively) compared to the DMSO control.

On the other hand, *ist1* gene expression was significantly decreased in the *A. dahurica* 1.5 µg/mL, 6 µg/mL, and 9 µg/mL, and *A. pubescens* 1.5 µg/mL groups (Figure 4D; *p* < 0.001, *p* < 0.001, *p* < 0.001, *p* < 0.01, respectively) compared to the DMSO control. Compared to the diazepam-treated group, significant reductions in *ist1* gene expression were observed in all treatment groups (*A. dahurica* 1.5 µg/mL, 6 µg/mL, 9 µg/mL, and *A. pubescens* 1.5 µg/mL, 6 µg/mL, 9 µg/mL; *p* < 0.001 for all except *A. pubescens* 9 µg/mL which was *p* < 0.01).

For *avp* gene expression (Figure 4E), significant increases were detected in the diazepam-treated, *A. pubescens* 6 µg/mL, and 9 µg/mL groups (*p* < 0.05, *p* < 0.001, *p* < 0.001, respectively) compared to the DMSO control.

Conversely, significant reductions were observed in the *A. dahurica* 1.5 µg/mL, 6 µg/mL, and 9 µg/mL groups (*p* < 0.001, *p* < 0.01, *p* < 0.001, respectively). Compared to the diazepam-treated group, reductions in *avp* gene expression were observed in the *A. dahurica* 1.5 µg/mL, 6 µg/mL, and 9 µg/mL groups (*p* < 0.001, *p* < 0.01, *p* < 0.001, respectively), with a slight reduction in the *A. pubescens* 1.5 µg/mL group (*p* < 0.05).

Finally, the *nr3c1* gene expression (Figure 4F) was significantly decreased in the diazepam-treated group compared to the DMSO control group (*p* < 0.001). However, significant increases in *nr3c1* gene expression were observed in the *A. dahurica* 1.5 µg/mL, 6 µg/mL, and 9 µg/mL groups (*p* < 0.001 for all). Compared to the diazepam-treated group, significant increases in *nr3c1* expression were found in all treatment groups (*A. dahurica* 1.5 µg/mL, 6 µg/mL, 9 µg/mL, *A. pubescens* 1.5 µg/mL, 6 µg/mL, 9 µg/mL; *p* < 0.001 for all).

For *nr3c2* gene expression (Figure 4G), significant increases were observed in the *A. dahurica* 1.5 µg/mL, *A. pubescens* 1.5 µg/mL, 6 µg/mL, and 9 µg/mL groups compared to the DMSO control (*p* < 0.001 for all). In contrast, significant decreases were noted in the *A. dahurica* 6 µg/mL and 9 µg/mL groups (*p* < 0.001 for both). Furthermore, compared to the diazepam-treated group, a significant increase in *nr3c2* expression was observed in the *A. pubescens* 1.5 µg/mL, 6 µg/mL, and 9 µg/mL groups (*p* < 0.01, *p* < 0.01, *p* < 0.001, respectively), while expression was significantly reduced in the *A. dahurica* 6 µg/mL and 9 µg/mL groups (*p* < 0.01, *p* < 0.001, respectively).

## 3. Discussion

*A. pubescens* and *A. dahurica* are two plants with well-documented pharmacological properties, including anti-inflammatory, antimicrobial, and free radical scavenging activities [11]. Additionally, *A. pubescens* has been reported to reduce pain and fever, aid in the treatment of osteoporosis and neurodegenerative diseases, and inhibit apoptosis [14]. In contrast, *A. dahurica* is known for its broad therapeutic effects, including anti-inflammatory and pain-relieving properties, enhancement of angiogenesis, and its potential role in the treatment of depression and hyperlipidemia [25]. Furthermore, *A. dahurica* has demonstrated the ability to reduce tumor growth, modulate immune responses, and alleviate anxiety-like behaviors [26]. Due to these diverse pharmacological activities, both plants hold significant therapeutic potential, particularly in the context of stress and anxiety disorders.

Given the promising effects of these plants, this study aimed to investigate their anxiolytic potential in *Danio rerio* under stress conditions induced by light exposure. The zebrafish model has been increasingly utilized for studying anxiety and stress due to its well-characterized behavioral responses to environmental stressors, such as alterations in locomotion and social interactions [18]. In this study, zebrafish were exposed to a light-induced stress stimulus after treatment with diazepam (as a positive control) and plant extracts from *A. dahurica* and *Angelica pubescens*. DMSO was used as a negative control.

Zebrafish exhibit characteristic behavioral changes under stress, such as decreased activity, increased immobility, and altered swimming patterns, which are commonly associated with anxiety and stress. Diazepam, a well-known anxiolytic, induces sedation, drowsiness, and reduced swimming activity in zebrafish, providing evidence for its anxiolytic effect [27]. In this study, we measured the mRNA expression of several genes, including *hcrt*, *crhr1*, *ccka*, *ist1*, *avp*, *nr3c1*, and *nr3c2*, which were selected based on their relevance to anxiety, sleep–wake regulation, and stress responses. These genes are integral to the physiological mechanisms underlying anxiety and stress, making them valuable biomarkers for investigating the effects of anxiolytic treatments in zebrafish.

### 3.1. Toxicity Studies

Toxicological assessment is one of the most crucial stages of preclinical (and clinical) studies. On one hand, it allows for the determination of which organs/tissues can be damaged after exposure to a given substance (or a combination of substances), and on the other hand, it allows for the selection of a safe dose of a given substance for further testing. Usually, rodents such as white rats and mice, rabbits, guinea pigs, hamsters, and, less often, dogs, are used as research models in toxicological studies. However, the suffering of animals subjected to toxicological tests raises controversy and objection. Therefore, alternative methods are being searched for, including studies on cell lines or in zebrafish models. The use of *Danio rerio* as a research model allows for faster and more economical toxicity tests when compared to the ones carried out in rats or mice. Embryos of *Danio rerio* are susceptible and sensitive to toxins diluted in water. Therefore, the embryotoxicity test allows for an assessment of the toxic effects of a solution of a given substance in which the zebrafish embryos/larvae are immersed [28]. In addition, *Danio rerio* has a transparent body, small size, rapid development, and low requirements in relation to breeding, feeding, and keeping [29]. Even though the zebrafish model is particularly useful for testing developmental toxicity [30], a good correlation between outcomes obtained in the embryotoxicity test applied in our study and results obtained in acute toxicity tests carried out in mature animals was recorded [15]. In the presented project, the assessment of the toxicity of the tested *Angelica* extracts was made on the basis of monitoring any abnormalities in the development as well as in the embryonic/larval growth of *Danio rerio.* Mortality, the presence of oedemas, body bending, abnormalities in the developing organs (eyes, otoliths, fins, yolk, swimbladder, and notochord), and pigmentation were observed meticulously under a microscope. One experiment was conducted for 5 days.

The results of toxicity studies revealed that zebrafish exposure to solutions of *Angelica dahuria* extract resulted in a similar mortality to the one observed for control groups, with the LD50 above 200 µg/mL. However, only the lowest tested concentration of the extract was safe for the development of *Danio rerio* embryos/larvae since oedemas and/or malformations (particularly in the heart areas) were observed in almost all subjects exposed to the concentration range of 200–300 µg/mL. At the lowest tested concentrations of *Angelica dahuria* extract (i.e., 15 and 30 µg/mL), the development of embryos/larvae was normal in relation to eye formation, otoliths, and pigmentation.

Solutions of *A. pubescens* extract caused significantly higher mortality of zebrafish embryos and larvae than solutions of *Angelica dahuria* extracts. The mortality rate amongst subjects incubated in concentrations of 200 and 100 µg/mL were 100% and 22.5%, respectively, after a 24 h exposure. It was generally in line with the findings by Yang and colleagues [31]. After a longer incubation period, a further increase in mortality was observed, particularly in the 100 µg/mL solution, in which after a 96 h exposure, the percentage of dead individuals was 75%. Swellings next to the heart area and enlarged, distorted yolks observed in zebrafish embryos/larvae incubated in solutions of *A. pubescens* extract were much different in appearance from the same type of deformations detected in embryos/larvae exposed to solutions of *Angelica dahuria* extract. The malformations were larger and caused the fish’s body to curve or bend. Moreover, at the highest concentrations in which the larvae survived, i.e., 50 and 100 µg/mL, the body of *Danio rerio* was smaller and thickened as if it was swollen. The pigmentation of subjects that were developing in solutions of *A. pubescens* was different from the pigmentation of control larvae swimming in E3 medium or 1% solution of DMSO. It is highly probable that both *Angelica* extracts may have some cardiotoxic potential since all tested concentrations slowed down the heart rate of zebrafish larvae after a 96 h incubation. However, to confirm such a toxic effect on the hearts of higher animals (and humans), further studies have to be carried out.

Based on the results obtained, the dose-dependent toxicity of *Angelica* extracts was observed. With increased concentrations of the extracts, the degree of toxicity was higher; exposure to higher doses of the extracts was accompanied by an increased number of oedemas along with their larger size. The same relation was observed for exposure time; a longer incubation was accompanied by an increased number of edemas along with their larger size. The latter correlation might have been influenced, at least to some extent, by the presence of chorion, which, during the initial stage of the development of an embryo, serves as a protective barrier against external factors. A chorion could have limited, to some extent, the direct contact of a given extract with a developing embryo. However, once a larva hatched, it was more exposed to the toxic effects of tested extracts.

It should be mentioned that the observed toxicity of *Angelica* extracts was not due to the addition of DMSO since the development of zebrafish embryos/larvae incubated in 1% solution was similar to the one observed for subjects living in the E3 medium. Based on the obtained results, we can assume that the tested extracts did not significantly reduce or delay the hatching process. Exposure to solutions of *A. pubescens* extract may even accelerate leaving the chorions by larvae, since despite visible deformations, the percentage of hatched individuals incubated in solutions of *A. pubescens* extract for 72 h was between 32 and 60.5 (for the control group living in E3 medium it was 20.5%). After a 96 h exposure to the tested solutions (E3 medium, 1% solution of DMSO, solutions of *A. dahurica* and *A. pubescens* extracts), all larvae had already left their chorions, which was consistent with the OECD Guidelines [32].

### 3.2. Hcrt Gene Expression and Its Behavioral Implications

The results of this study revealed a statistically significant increase in the mRNA expression of the *hcrt* gene in the diazepam-treated group compared to the DMSO control. The *hcrt* gene encodes a precursor protein, which is cleaved into two forms of orexin: orexin-A (*hcrt-1*) and orexin-B (*hcrt-2*). These neuropeptides play a crucial role in regulating arousal, sleep–wake states, stress behaviors, feeding, and reward [33,34]. In zebrafish, one receptor corresponding to the HCRT-2 receptor in mammals has been identified [20,35]. Increased expression of the *hcrt* gene in zebrafish larvae has been shown to cause prolonged wakefulness and fragmented sleep. Additionally, administration of HCRT-1 to adult zebrafish has been reported to increase food intake and mobility [36]. These effects have also been observed in various fish species, where HCRT-1 significantly influenced locomotion and feeding behaviors [35,37]. In the current study, the increase in *hcrt* gene expression following diazepam administration in zebrafish larvae was unexpected, given that diazepam is known for its anxiolytic properties and generally produces sedative effects. However, our findings show an increase in wakefulness, reflected in elevated *hcrt* gene expression, which may be explained by the rapid onset and relatively short duration of action of diazepam. The study was conducted for 90 min after a 30-minute incubation period, and it is possible that the effects of diazepam were not fully manifested within this timeframe. These findings could be attributed to the short duration of the experiment, wherein the sedative effects of diazepam might not have fully manifested due to its rapid onset and relatively brief action. Furthermore, since the study was performed on cultured zebrafish rather than wild-type specimens, there may be strain-specific responses. Another possible explanation is that diazepam may act as an hcrt receptor antagonist, but this hypothesis requires further investigation. In the case of the plant extracts, we observed a slight increase in *hcrt* gene expression across all concentrations of *A. pubescens* compared to the DMSO control group. However, no significant changes were noted in the *A. dahurica* 1.5 µg/mL and 6 µg/mL groups. The only group showing a decrease in *hcrt* expression relative to the control was the highest concentration of *A. dahurica* (9 µg/mL), indicating a potential sedative effect at this dose. Interestingly, across all plant extract-treated groups, *hcrt* gene expression was lower compared to the diazepam-treated group, further supporting the hypothesis that diazepam’s effects may be related to its duration of action or dosage. Additionally, strain-specific responses in cultured zebrafish may contribute to the observed results, as genetic and environmental factors can influence drug responses. The possibility of diazepam acting as an hcrt receptor antagonist also warrants further investigation. Interestingly, the plant extracts demonstrated a dose-dependent effect on *hcrt* expression, with higher concentrations of *A. dahurica* showing a decrease in expression, suggesting a potential sedative effect. These results underline the complex interplay between pharmacological agents and orexin signaling pathways, emphasizing the need for further studies to delineate these interactions. The increase in *hcrt* gene expression by diazepam, despite its sedative effects, is consistent with recent studies suggesting that orexinergic systems are involved in the modulation of benzodiazepine actions, including wakefulness and arousal [38,39]. Similarly, plant extracts such as *A. pubescens* and *A. dahurica* have shown dose-dependent effects on hcrt expression, which may reflect their potential impact on orexin signaling and anxiety regulation [40]. A more systematic comparison with established anxiolytic drugs could help to clarify the role of orexin in the pharmacological effects of these natural compounds in the future [39,41].

### 3.3. Modulation of crhr1 Gene Expression and Anxiety

Corticotropin-releasing hormone (CRH) and its receptor, CRHR1, are known to be involved in the regulation of anxiety. Studies have shown that a deficiency of *crhr1* results in reduced anxiety-like behaviors, while elevated *crhr1* expression is linked to chronic stress and anxiety disorders [42]. The CRH/CRHR1 system plays a key role in the hypothalamic–pituitary–adrenal (HPA) axis, with CRH acting on the anterior pituitary to stimulate the release of adrenocorticotropic hormone (ACTH), which in turn affects the adrenal glands. In the basolateral amygdala (BLA), CRHR1 plays a pivotal role in mediating anxiety-like behaviors. Overexpression of *crhr1* is associated with heightened anxiety and stress reactivity, while its downregulation can reduce anxiety-like behaviors. The study by Savarese and Lasek (2018) highlights the importance of *crhr1* as a molecular target in the development of therapeutic strategies for anxiety disorders [43]. In the present study, we observed a significant reduction in *crhr1* gene expression in all experimental groups compared to the DMSO control, with the most pronounced decrease seen in the *A. dahurica* 1.5 µg/mL group. This reduction in *crhr1* expression was more substantial than in the diazepam group, which supports the hypothesis that the plant extracts may possess anxiolytic effects similar to diazepam. The *A. pubescens* groups showed results that were comparable to those of the diazepam-treated group, although there was a slight increase in *crhr1* expression in the 1.5 µg/mL and 9 µg/mL concentrations compared to diazepam. These results further suggest that both the plant extracts and diazepam lead to reduced *crhr1* expression, a marker of reduced anxiety. The substantial reduction in *crhr1* expression observed with both diazepam and plant extracts highlights their potential utility in managing anxiety disorders. The findings also suggest that the plant extracts could serve as alternative therapeutic agents, offering a comparable anxiolytic profile to diazepam but with potentially fewer side effects. The modulation of *crhr1* gene expression is crucial for understanding the regulation of anxiety. The CRH/CRHR1 system is involved in stress responses, and an imbalance in CRHR1 expression can lead to heightened anxiety and stress reactivity [43]. In our study, we observed that both *A. dahurica* and *A. pubescens* led to a significant reduction in *crhr1* expression, which was more pronounced in the *A. dahurica* group compared to the diazepam group. This suggests that these plant extracts may exert anxiolytic effects comparable to diazepam but potentially with a different mechanism or fewer side effects [44].

### 3.4. Ccka Gene Expression and Dose-Dependent Effects

The *CCKA* gene, which is involved in dopamine regulation and anxiety-related behaviors, was also examined in this study. CCKA is a neuropeptide receptor that plays a role in the regulation of satiety, dopamine-related behaviors, and anxiety. Previous studies in rats have shown that CCKA receptor agonists induce anxiogenic effects, whereas antagonists exert anxiolytic effects [23,45]. Studies have shown that *CCK* gene variants may be associated with different anxiety phenotypes, and CCKAR may play a role in the development of panic comorbid with bipolar disorder [46]. Administration of the tetrapeptide cholecystokinin (CCK-4) has been shown to induce panic attacks in both patients with panic disorder and healthy volunteers, suggesting a direct effect of CCK on the mechanisms of anxiety [47]. There are also unresolved issues regarding the involvement of brain CCK in the pathogenesis of anxiety and panic disorders, including therapeutic trials using CCK2 receptor antagonists [48]. In our study, *ccka* gene expression was reduced in the diazepam and *A. dahurica* 1.5 µg/mL groups compared to the DMSO control, suggesting that both diazepam and *A. dahurica* may act as *ccka* receptor antagonists, inducing anxiolytic effects. However, the *A. pubescens* 6 µg/mL and 9 µg/mL groups showed increased *ccka* gene expression, which might indicate different mechanisms of action for the plant extracts at higher doses. In our study, we observed a reduction in ccka gene expression in the diazepam and *A. dahurica* (1.5 µg/mL) treatment groups, suggesting that both may act as CCKA receptor antagonists, similar to the anxiolytic effects of diazepam. Notably, higher concentrations of *A. pubescens* (6 µg/mL and 9 µg/mL) resulted in increased ccka expression, indicating that different doses of plant extracts could activate distinct pathways, with higher doses potentially leading to anxiogenic effects, as reported in previous studies [9]. It is also important to note that other concentrations of *A. dahurica* and *A. pubescens* did not significantly alter ccka gene expression compared to the control group, highlighting the potential dose-dependent effects of these plant extracts. These findings suggest that while lower concentrations of the plant extracts may mimic the anxiolytic effects of diazepam, higher doses may activate different pathways. This underscores the importance of optimizing dosage in therapeutic applications.

### 3.5. Ist1 Gene Expression and Cognitive Implications

The IST1 gene (associated with ESCRT-III) plays a key role in nervous system functioning, participating in processes related to endosomal sorting and intracellular transport. Studies suggest that improper regulation of IST1 can lead to disorders in autophagy [49]. Recent studies have shown that IST1 is necessary for efficient cell division during cytokinesis and participates in the regulation of intracellular vesicle formation [50,51,52]. In our study, a significant increase in *ist1* expression was observed in the diazepam-treated group, suggesting its potential influence on synaptic plasticity and cognitive function. A similar increase in expression was noted at the highest concentration of *A. pubescens* (9 µg/mL), which may indicate a potential cognitive-enhancing effect of this plant extract at this dose. These results are unexpected, given diazepam’s primary use as an anxiolytic and sedative, and further investigation is needed to confirm whether *A. pubescens* can improve cognitive function in zebrafish larvae.

### 3.6. Avp Gene Expression and Stress Regulation

Arginine-8-vasopressin (AVP) is a neuropeptide that plays a key role in regulating the stress response [53]. It is released in response to stressors and influences behavior and neuroendocrine regulation through the activation of AVP receptors. Studies have shown that changes in *AVP* gene expression can lead to stress-related disorders such as anxiety and depression [54]. Additionally, animal studies suggest that the blockade of AVP receptors can reduce stress responses and improve memory, underscoring the central role of AVP in regulating the stress response. Finally, *avp* (arginine vasopressin) is another gene linked to anxiety and stress behaviors. Increased expression of *Avp* is associated with anxiety-like behaviors, and the presence of *avp* in the central amygdala has been shown to induce anxiety in animal models [55]. Pharmacological studies have shown that *A. dahurica* root and its active ingredients exhibit various biological properties, including anti-inflammatory, anticancer, antioxidant, analgesic, antiviral, and antimicrobial effects, effects on the cardiovascular system, and neuroprotective effects. [17,25,56]. In our study, *avp* gene expression was increased in the diazepam-treated group and at the 6 µg/mL and 9 µg/mL concentrations of *A. pubescens*, suggesting that both diazepam and higher doses of *A. pubescens* may exert an agonistic effect on AVP receptors. In contrast, all groups treated with *A. dahurica* showed reduced *avp* gene expression, suggesting a potential anxiolytic effect. These results are in line with previous studies demonstrating that plant extracts may modulate stress responses in different ways [57]. Thus, the plant extracts examined in our study may modulate anxiety through various mechanisms depending on the dose and type of treatment.

### 3.7. Nr3c1 and nr3c2 Genes Expression: Insights into HPA Axis Regulation

The *nr3c1* gene encodes the glucocorticoid receptor, which binds cortisol, the stress hormone produced by the adrenal glands [58]. Methylation of this gene is associated with reduced expression of the gene, resulting in a decrease in the number of glucocorticoid receptors [59,60]. This process is observed in various conditions, such as depression, personality disorders, and anxiety disorders. On the other hand, demethylation of *nr3c1* is associated with diseases such as obesity. Previous studies using zebrafish have shown that exposure to stress-inducing stimuli activates the hypothalamic–pituitary–adrenal (HPA) axis, resulting in increased motor activity. Recent studies have further elucidated the role of *nr3c1* methylation in stress-related disorders [59]. Hypermethylation of NR3C1 has been shown to be associated with negative outcomes in children, including greater emotional lability, higher levels of ego control, more externalizing behavior, and greater depressive symptoms. These results suggest that methylation of NR3C1 may contribute to the development of psychopathology in children. DNA methylation studies of the glucocorticoid receptor gene in post-traumatic stress disorder (PTSD) and immunity have highlighted the role of *NR3C1* DNA methylation in HPA axis dysregulation in major depressive disorder (MDD), indicating that epigenetic modifications of *NR3C1* may affect stress reactivity and contribute to the pathophysiology of MDD [61]. Epigenetic changes to this gene have been shown to affect stress responsiveness and immunity. This process is observed in various conditions, such as depression, personality disorders, and anxiety disorders. Conversely, demethylation of *nr3c1* is linked to diseases such as obesity [22,62]. In previous studies using zebrafish, exposure to stress-inducing stimuli has been shown to activate the hypothalamic–pituitary–adrenal (HPA) axis, resulting in increased motor activity [40]. Interestingly, the results obtained in this study were not in line with our expectations. Diazepam, a well-known anxiolytic, caused a decrease in *nr3c1* gene expression. This suggests that diazepam may have induced methylation of the *nr3c1* gene, which, as previously mentioned, is associated with conditions like depression and anxiety disorders. One potential explanation for this unexpected result is the environmental context in which the zebrafish were exposed to prior stress, possibly leading to dysfunction in the HPA axis. Additionally, the dose of diazepam used in the study might have been too low to produce the expected anxiolytic effect, while the light stimulus during the experiment could have further activated the HPA axis, contributing to the observed decrease in gene expression. These factors combined may have altered the expected response to diazepam. In contrast to the diazepam-treated and control groups, the plant extracts derived from *A. dahurica* led to significantly increased *nr3c1* gene expression. This result suggests that the *A. dahurica* extracts may have conferred resistance to glucocorticosteroids. Such an effect would imply that *A. dahurica* has a potential role in modulating the stress response and could contribute to the reduction in stress levels. This finding further supports the hypothesis that *A. dahurica* may be useful in treating anxiety and sleep–wakefulness disorders, as it seems to counteract the negative effects of stress through regulation of the HPA axis. The *nr3c2* gene encodes the mineralocorticoid receptor, which is involved in the regulation of HPA axis activity and is associated with anxiety and stress behaviors [63]. An increase in *nr3c2* gene expression has been observed in zebrafish subjected to inhibited avoidance tests, indicating its role in stress responses [64]. Furthermore, a study on infants whose mothers exhibited depressive symptoms during pregnancy showed that *nr3c2* methylation was linked to decreased cortisol reactivity, which further suggests the gene’s relevance in stress regulation and anxiety [65]. In the present study, no significant changes in *nr3c2* gene expression were observed in the diazepam-treated group compared to the DMSO control, suggesting that diazepam did not impact the methylation status of this gene under the conditions of this experiment. However, the plant extracts of *A. dahurica* and *A. pubescens* displayed varying effects on *nr3c2* gene expression. The 1.5 µg/mL concentrations of both *A. dahurica* and *A. pubescens* showed an increase in *nr3c2* gene expression, which suggests that demethylation of the gene may have occurred. This demethylation has likely resulted in an increase in cortisol reactivity and a reduction in anxiety levels. This is consistent with the anxiolytic potential of these plant extracts, particularly at lower concentrations. Interestingly, higher concentrations of *A. dahurica* (6 µg/mL and 9 µg/mL) led to a decrease in *nr3c2* gene expression, which suggests methylation of the gene and a reduction in mineralocorticoid receptor activity. This effect could indicate an increase in stress levels and potentially an anxiogenic response. The observed result may be due to the dose-dependent nature of the plant extracts, where lower concentrations promote a reduction in anxiety, while higher concentrations may induce a stress response. This finding aligns with the complex interactions between plant compounds and stress-related pathways and further emphasizes the importance of dosage in determining the pharmacological effects of herbal treatments. These results suggest that *A. dahurica* has a dose-dependent effect on the regulation of the HPA axis, with lower concentrations leading to reduced anxiety and higher concentrations potentially exacerbating stress levels. Conversely, *A. pubescens* exhibited a more consistent effect across different concentrations, indicating its potential as a therapeutic agent for anxiety and stress-related disorders. These findings emphasize the importance of dosage in determining the therapeutic efficacy of *A. dahurica* and *Angelica pubescens.* The differential effects on *nr3c1* and *nr3c2* expression also suggest that these plant extracts modulate the HPA axis through distinct pathways, offering insights into their potential as treatments for stress and anxiety-related disorders. However, further studies are necessary to better understand the underlying mechanisms and the precise dosage required to achieve the desired anxiolytic effect.

In our study, we noticed that *A. dahurica* increased the expression of genes related to stress response, including nr3c1, which is a key element in the regulation of the HPA axis. This is consistent with the results of previous studies, which suggest that increased expression of *nr3c1* may lead to a better response to stress and reduced anxiety symptoms. Liu and Nusslock (2018) [58] noted and demonstrated the connection between GR, HPa, and the gene—they showed that excessive methylation of the NR3C1 gene promoter region weakens the expression of GR, which disrupts the negative feedback mechanism in the HPA axis, leading to dysregulation of the neuroendocrine response to stress [17]. Such effects suggest that *A. dahurica* may have therapeutic potential in the treatment of anxiety and stress disorders, acting through modulation of stress response at the molecular level. In another study, Zhao et al. (2022) [56] showed that *A. dahurica* has strong anti-inflammatory and antioxidant effects, thanks to crude extracts and isolated compounds such as IMP, which modulate the expression of pro-inflammatory mediators (NF-κB, iNOS, COX-2, TNF-α) and improve resistance to oxidative stress. *A. dahurica* may exert adaptogenic properties by influencing the expression of genes, such as nr3c1, involved in the regulation of the HPA axis. The expression of *nr3c1* in response to stress suggests the potential of this plant extract to stabilize the body’s response to stress, which may be promising in the treatment of disorders related to chronic stress and anxiety.

## 4. Materials and Methods

### 4.1. Plant Material and Extraction

The plant material used in this study consisted of 1 g of dried and powdered roots of *A. dahurica* (Hoffm.) Benth. & Hook. f. ex Franch. & Sav. and *A. pubescens* Maxim. These raw materials were transferred to a 10 mL extraction cell and extracted using methanol through pressurized liquid extraction (ASE). Each extraction cycle lasted 10 min, and the plant material was extracted three times, each time with a fresh portion of methanol. The extraction was performed at 80 °C and a pressure of up to 110 atm.

Following the extraction, the obtained extracts were collected in a 50 mL volumetric flask and made up to volume with the same solvent used for extraction. The extracts were then stored in a refrigerator.

To remove the solvent, a vacuum evaporator under reduced pressure was used. A total of 50 mL of the extract was taken, and distillation was carried out until the solvent completely evaporated. After vacuum distillation, the final weights of the extracts were as follows: *A. dahurica* (0.37435 g) and *A. pubescens* (0.43118 g). A portion of each extract (approximately 0.2 g) was then dissolved in DMSO for further analysis.

### 4.2. HPLC/ESI-QTOF-MS

The purified samples were qualitatively analyzed by an HPLC/ESI-QTOF-MS system in positive ion mode using a 6530B Accurate-mass-QTOF-MS (Agilent Technologies, Inc., Santa Clara, CA, USA) mass spectrometer with an ESI-Jet Stream ion source. The Agilent 1260 chromatograph was equipped with a DAD detector, autosampler, binary gradient pump, and column oven. Solvent gradient: water with 0.1% formic acid (solvent A) and acetonitrile with 0.1% formic acid (solvent B) were used as mobile phases. The stationary phase was a Phenomenex Gemini C18 3 µm 100 × 2 mm chromatography column (Phenomenex, Torrance, CA, USA). The following gradient procedure was used: 0–10 min, 40% B; 10–30 min, 40–80% B; 30–35 min, 80–90% (B); the post time was 10 min. The total analysis time was 45 min, with a stable flow rate of 0.200 mL/min. The injection volume for extracts was 10 μL. ESI-QToF-MS analysis was performed according to the following ion source parameters: dual spray jet stream ESI, positive and negative ion mode, gas (N2) flow rate 12 L/min; nebulizer pressure: 35 psig; evaporator temperature: 300 °C; *m*/*z* range 100–1000 mass units, with Auto MS/MS acquisition mode, collision-induced dissociation (CID): 10 and 30 eV with MS scan rate 1 spectrum per s, 2 spectra per cycle, Skimmer: 65 V, fragmentor: 140 V and octopole RF peak: 750 V.

### 4.3. Animals

Zebrafish behavioral tests were performed at the Center for Experimental Medicine in Lublin, Poland. Zebrafish (*Danio rerio*) were maintained at 26 ± 1 °C with a light/dark cycle of 14 h of light and 10 h of dark. The water was continuously filtered and disinfected with ultraviolet. One female and one male were transferred to the breeding tank, spawning was induced using light. The obtained eggs were collected, and after rinsing, they were placed in Petri dishes in embryo solution (E3 medium), which consists of 5 mM NaCl, 0.17 mM KCl, 0.33 mM CaCl_2_, 0.33 mM MgSO_4_). The groups participating in the experiment are presented in Table 4.

#### 4.3.1. Toxicity Studies

After a 180 min incubation at 28.5 °C (±0.3 °C), fertilized eggs were selected for the experiment. Subjects that did not show any signs of coagulation or deformation were used.

#### 4.3.2. Behavioral and Molecular Studies

Embryos were matured in an incubator at 28.5 °C under a light/dark cycle where light was 14 h and dark was 10 h. Zebrafish larvae at the 5 days post fertilization (dpf) stage were used. Since the used zebrafish larvae were not older than 5 dpf, the approval of the Local Ethics Committee for Animal Experiments was not required.

#### 4.3.3. Experimental Design

The fish embryo toxicity test was carried out according to the OECD Guidelines [32] with modifications. The tested extracts were at first dissolved in DMSO and, after that, diluted in E3 medium. The final concentration of DMSO in applied solutions was 1% *v*/*v*, which is deemed safe for the development of *Danio rerio* [66]. To confirm that this concentration of DMSO did not falsify the obtained results, the respective control group was used that was exposed to a 1% solution of DMSO. Three hpf selected eggs were transferred to 24-well plates (1 embryo/well). Eggs were exposed to different solutions (20 subjects/group): (I) E3 medium (control group); (II) 1% *v*/*v* DMSO (in E3 medium); (III–VII) solutions of *A. dahurica* extract (200, 100, 50, 30, 15 µg/mL); (VIII–XII) solutions of *A. pubescens* extract (200, 100, 50, 30, 15 µg/mL). The concentration ranges used in the fish embryo toxicity test were selected based on previous studies and preliminary observations. Our goal was to cover a range of subtoxic to potentially harmful levels to assess both therapeutic effects and toxicity. Lower concentrations were selected to assess potential benefits, while higher concentrations tested toxicity thresholds. This approach allowed us to observe both positive and negative effects on developing zebrafish embryos. The selected ranges were also confirmed by similar studies in the literature and our knowledge of the properties of plant extracts. Each experiment was performed in two replicates. Thus, the total number of subjects per group was 40. After the first day of incubation of the embryos in the tested solutions, only the presence of coagulation (indicating the death of embryos) was assessed, whereas after 48, 72, and 96 hpf, observations of embryos were made in order to identify lethal (coagulation and absence of heartbeat) and sublethal (malformation of the head, eyes, tail or heart, bent spine, deformity of the yolk, reduction in pigmentation and delay or reduction in hatching) endpoints [67]. After 96 h post-fertilization (hpf), zebrafish larvae were carefully examined to assess their heart rate. Heart rate measurements were taken using an automated video-tracking system connected to a microscope. Each larva was placed in a 24-well plate, and the heart rate was monitored using specialized software to record and analyze the frequency of heartbeats. The larvae were acclimated for a period of 10 min under controlled light conditions prior to heart rate measurement. Following this, the heartbeats were recorded for 5 min per larva, ensuring consistent and accurate readings. Heart rate was measured under normal conditions, and the values were used to assess the potential effects of the plant extracts and diazepam on the autonomic nervous system. Changes in heart rate were used as an indicator of the physiological response to the experimental treatments, with a significant reduction or increase potentially signaling toxicity or a stress response. The experiment lasted for 5 days. After that, larvae were disposed of according to an in-house procedure. For 5 days, embryos/larvae were incubated at the following ambient conditions: temperature of 28.5 ± 0.3 °C, a 14/10 h light/dark cycle.

The anxiolytic properties of the obtained plant extracts were tested on 5 dpf zebrafish larvae. Larvae were transferred individually into wells of 24-well plates. Before the experiment, solutions were prepared: plant extracts dissolved in DMSO were diluted with E3 medium to obtain concentrations of 1.5, 6, and 9 µg/mL. Solutions were added at 1.5 mL to each well. Diazepam was dissolved in DMSO and diluted to a concentration of 10 µM using E3 solution and then added 1.5 mL to each well. The control group was a 1% DMSO solution added 1.5 mL to each well. Zebrafish larvae were incubated for 30 min before testing with DMSO (1%), diazepam (10 µM), and plant extract (1.5, 6, 9 µg/mL) solutions at 28.5 °C in the dark.

After the incubation period, plates were placed in a multi-well plate holder inside the automated video recording bench station (View Point, Lyon, France). The inside and outside of the wells were marked to examine larval activity. Initially, zebrafish were acclimated for 10 min under light conditions. Then, spontaneous activity was studied for 85 min: 40 min of continuous light followed by 3 light/dark cycles (10 min light/5 min dark).

Anxiety-inducing activity was defined as an increase in larval residence time in the inner part of the well. After the experiment was completed, the larvae were immobilized in ice-cold water. Each sample that was obtained represented 24 larvae. After the excess water was removed, the samples were frozen.

### 4.4. Real-Time Polymerase Chain Reaction (qPCR)

Total RNA was isolated from 15 *Danio rerio* larvae by TRIzol reagent (Invitrogen, Waltham, MA, USA) according to manufacturer protocol. The spectrophotometer (Maestrogen, Hsinchu, Taiwan) was used to quantify RNA content and quality. Only high-purity RNA (RNA: A260/280 > 1.9) was normalized and used for further cDNA generation. cDNA synthesis was carried out with the NG dART RT-PCR kit (EURx, Gdańsk, Poland) according to the manufacturer’s instructions. QPCR assays were performed using PowerUp SYBR Green Master Mix (Applied Biosystems, Foster, CA, USA) along with primers listed in Table 5 and normalized to rpl8, eef1a1l1, and actb1 genes. The genes’ expression levels were analyzed using the real-time PCR method and a 7500 fast real-time PCR system (Applied Biosystems, Foster City, CA, USA). Each reaction was carried out in triplicates. The reaction profile: 95 °C/2 min, 40 cycles: 95 °C/15 s, 57 °C/15 s, and 72 °C/1 min; melt curve 0.4 °C/s up to 97 °C. Results of gene expression alterations at the mRNA level were given as mean RQ ± SD. The concentration (µg RNA/mL) and A260/A280 ratio of samples are presented in Table 6.

### 4.5. Statistical Analysis

The program that was used for statistical analysis of the results obtained was GraphPad Prism 5 with one-way ANOVA extension with the post hoc test by Tukey (for behavioral and molecular studies) or with the post hoc test by Dunnett (for the heartbeat measurement). Statistically significant comparisons between the tested groups were present if the *p*-value was equal to or less than 0.05. The graphs showed the results obtained in the arithmetic mean and standard deviation (±SEM) form. For the analysis of toxicity studies, the zero-one scale was used (no mortality/no malformation = 0; mortality/malformation = 1; hatching = 1; no hatching = 0).

## 5. Conclusions

In conclusion, both *A. dahurica* and *A. pubescens* demonstrate potential for the treatment of anxiety and related disorders, particularly in modulating stress responses. *A. dahurica* exhibited relatively low toxicity, with promising effects at lower concentrations, suggesting its potential as a protective agent against stress. However, *A. pubescens* showed higher toxicity, especially at elevated doses, requiring careful dosage management. Both species affected genes related to the stress response and regulation of the HPA axis. However, their effects were dose-dependent, with lower concentrations exerting anxiolytic effects and higher concentrations potentially increasing stress. Unexpected effects were also observed with diazepam, highlighting the complexity of gene expression modulation under varying conditions. Overall, while these plant-based compounds are promising alternatives or adjuncts in anxiety therapy, further studies are needed to optimize dosages, explore long-term effects, and fully understand the mechanisms of action. The lack of protein analysis for the genes modulated in this study remains a significant limitation that requires further investigation.

### Future Directions

Further studies are needed to elucidate the precise mechanisms of action for diazepam and plant extracts on orexin and vasopressin signaling, investigate the long-term effects of these treatments on behavior and gene expression, optimize dosages for therapeutic applications to maximize anxiolytic effects while minimizing potential stress-related responses and explore the effects of these compounds in wild-type zebrafish and other animal models to validate their broader applicability. Future studies should include protein expression analysis to confirm the functional implications of the observed genetic changes.

## Figures and Tables

**Figure 1 ijms-26-02884-f001:**
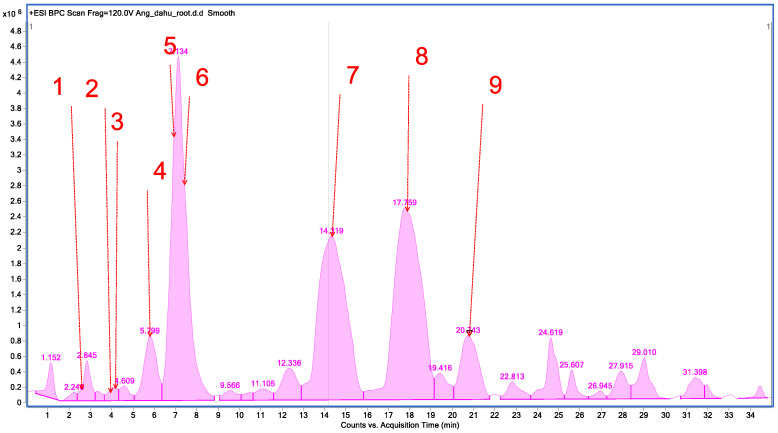
Base peak chromatogram of *A. dahurica* root L. methanolic extract by high-performance liquid chromatography–electrospray ionization–quadrupole time-of-flight–mass spectrometry (HPLC/ESI-QTOF-MS).

**Figure 2 ijms-26-02884-f002:**
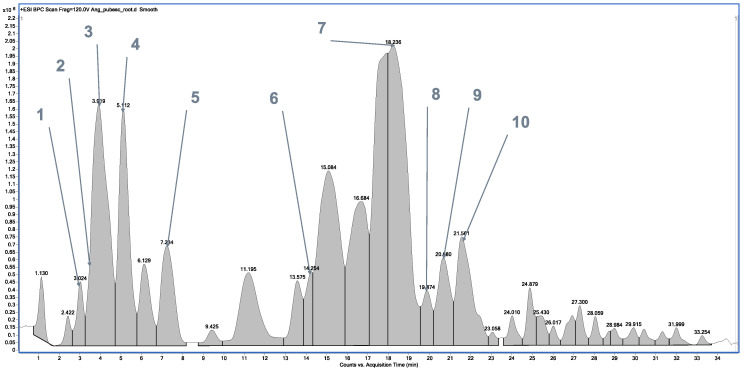
Base peak chromatogram of *A. pubescens* root L. methanolic extract by high-performance liquid chromatography–electrospray ionization–quadrupole-time of flight–mass spectrometry (HPLC/ESI-QTOF-MS).

**Figure 3 ijms-26-02884-f003:**
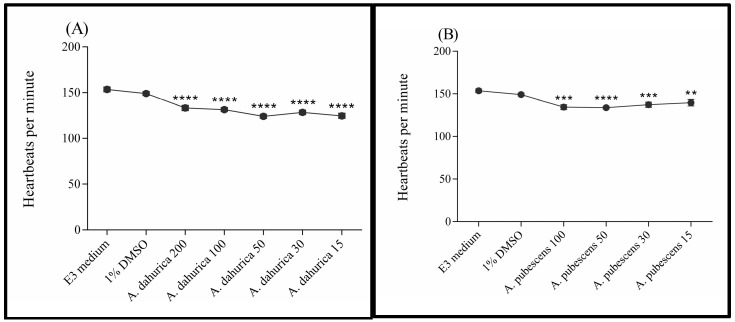
Activity of the zebrafish larvae heart after 96 h of exposure to *A. dahurica* (**A**) and *A. pubescens* (**B**). ** *p* < 0.01; *** *p* < 0.001; **** *p* < 0.0001 compared to DMSO control.

**Figure 4 ijms-26-02884-f004:**
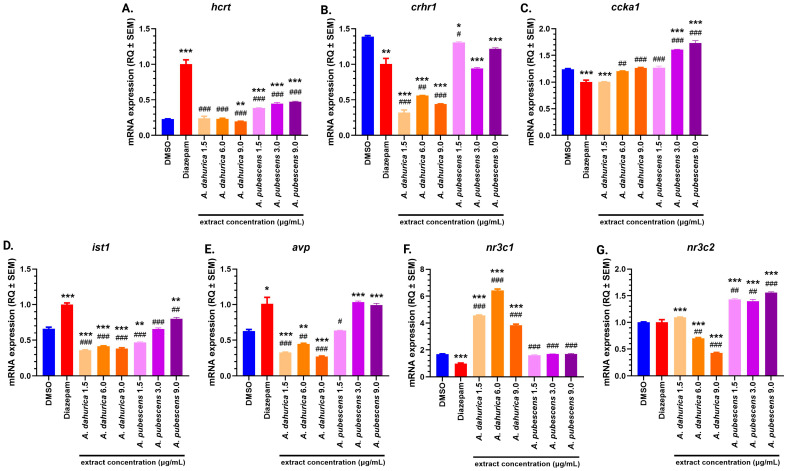
(**A**). Hypocretin neuropeptide precursor *(hcrt*); (**B**). corticotropin releasing hormone receptor 1 (*crhr1*); (**C**). cholecystokinin A receptor (*ccka1*); (**D**). factor associated with ESCRT-III (*ist1*); (**E**) arginine-8-vasopressin (*avp*); (**F**). glucocorticoid receptor (*nr3c1*); (**G**). mineralocorticoid receptor *(nr3c2*). The results are presented as mean ± SEM; * *p* < 0.05; ** *p* < 0.01; *** *p* < 0.001 compared to DMSO control; # *p* < 0.05; ## *p* < 0.01; ### *p* < 0.001 compared to diazepam control.

**Table 1 ijms-26-02884-t001:** Identification using ESI-QTOF-MS analysis of coumarins presents the methanolic extract of *A. dahurica* root.

No.	Tentatively Assignment	Formula	Retention Time	Molecular Ion [M + H]^+^	Molecular Ion Sodium Adduct [M + Na]^+^	Fragment Ion
1.	Oxypeucedanin hydrate	C_16_H_16_O_6_	2.682	305.0920	-	203.0443; 175.0517; 159.0511; 147.0503; 131.0572.
2.	Psoralene	C_11_H_6_O_3_	4.374	187.0323	-	143.0683; 131.0612; 115.0603; 103.0618.
3.	8-Methoxypsoralene	C_12_H_8_O_4_	4.483	217.0423	-	202.0372; 189.0630; 174.0414; 161.0688; 146.0434; 131.0556; 118.0470.
4.	Oxypeucedanin	C_16_H_14_O_5_	5.871	-	309.0796	224.0125.
5.	Byakangelicol	C_17_H_16_O_6_	6.885	317.0916	-	231.0265; 218.0191; 203.0316; 188.0104; 175.0383; 160.0143; 147.0389.
6.	Heraclenin	C_16_H_14_O_5_	7.602	287.0828	-	203.0258; 175.0300; 159.0351; 147.0348; 131.0438.
7.	Imperatorin	C_16_H_16_O_4_	14.866	271.0883	-	203.0291; 175.0370; 157.0256; 147.0437; 131.0457.
8.	Phellopterin	C_17_H_16_O_5_	17.759	301.0978	-	233.0418; 218.0204; 190.0239; 162.0312.
9.	Isoimperatorin	C_16_H_16_O_4_	20.743	271.0883	-	203.0317; 159.0429; 147.0424; 131.0495; 119.0464; 103.0535.

**Table 2 ijms-26-02884-t002:** Identification using ESI-QTOF-MS analysis of coumarins present in the methanolic extract of *A. pubescens* root.

No.	Tentatively Assignment	Formula	Retention Time	Molecular Ion [M + H]^+^	Molecular Ion Sodium Adduct [M + Na]^+^	Fragment Ion
1.	Nodakenetin	C_14_H_14_O_4_	2.871	247.0911	-	175.0259; 147.0333; 119.0492.
2.	Auraptenol	C_15_H_16_O_4_	3.491	261.1100	-	243.0938; 213.0476; 185.0543; 131.0457; 103.0497.
3.	Angelol G	C_20_H_24_O_7_	3.910	377.1552	-	277.1087; 259.0932; 231.0395; 219.0616; 205.0490; 191.0298; 175.0329; 160.0530.
4.	7-Methoxy-5-prenyloxycoumarin	C_15_H_16_O_4_	5.112	-	261.1042	189.0485; 131.0474; 103.0529.
5.	Columbianetin acetate	C_16_H_16_O_5_	7.206	289.0978	-	229.0834; 187.0388; 175.0393; 159.0415; 147.0440; 131.0489.
6.	Imperatorin	C_16_H_16_O_4_	14.087	271.0882	-	203.0339; 175.0350; 159.0396; 147.0416; 131.0444.
7.	Osthol	C_15_H_16_O_3_	18.236	245.1096	-	189.0480; 131.0478; 103.0532.
8.	Phellopterin	C_17_H_16_O_5_	19.882	301.0974	-	233.0317; 218.0089; 173.0150; 162.0222; 134.0329.
9.	Isoimperatorin	C_16_H_16_O_4_	20.808	271.0882	-	203.0326; 159.0444; 147.0422; 131.0467; 119.0479; 103.0537.

**Table 3 ijms-26-02884-t003:** Effects of exposure of *Danio rerio* embryos/larvae to solutions of *A. dahurica* and *A. pubescens* extracts. Studies were carried out on 40 embryos per group. Malformations and hatching were calculated as a percentage of living embryos/larvae at a given time point. DMSO (dimethyl sulfoxide), hpf (hours post fertilization), mortality: no mortality = 0, mortality = 1; malformations: no malformation = 0, malformation = 1; hatching: hatching = 1; no hatching = 0.

Solution		24 hpf	48 hpf			72 hpf			96 hpf		
		Mortality	Mortality	Malformations	Hatching	Mortality	Malformations	Hatching	Mortality	Malformations	Hatching
E3 medium		15%	15%	0	20.5%	15%	6%	100%	15%	6%	100%
DMSO	1%	15%	15%	3%	26.5%	15%	3%	100%	15%	3%	100%
*A. dahurica* extract	200 µg/mL	7.5%	7.5%	100%	8%	7.5%	100%	97%	7.5%	100%	100%
	100 µg/mL	22.5%	22.5%	100%	9.5%	22.5%	100%	96.5%	22.5%	100%	100%
	50 µg/mL	5%	5%	47%	10.5%	5%	100%	100%	5%	100%	100%
	30 µg/mL	15%	15%	0	11.5%	15%	67.5%	94%	15%	67.5%	100%
	15 µg/mL	15%	15%	0	11.5%	15%	3%	94%	15%	3%	100%
*A. pubescens* extract	200 µg/mL	100%	100%	-	-	100%	-	-	100%	-	-
	100 µg/mL	22.5%	22.5%	100%	32%	27.5%	100%	93%	75%	100%	100%
	50 µg/mL	7.5%	7.5%	100%	54%	10%	100%	100%	10%	100%	100%
	30 µg/mL	12.5%	12.5%	71%	60%	15%	73.5%	100%	15%	73.5%	100%
	15 µg/mL	17.5%	17.5%	0	60.5%	20%	0	100%	20%	0	100%

**Table 4 ijms-26-02884-t004:** The groups participating in the experiment.

1. DMSO control	100 µL DMSO + 9900 µL E3; 1.5 mL of 1% DMSO to each well
2. Diazepam control	5.7 µL diazepam + 100 µL DMSO + 9899.3 µL E3; 1.5 mL of solution to each well
3. *A. dahurica* 1.5 µg/mL	0.9 µL *A. dahurica* + 119.1 µL DMSO + 11,880 µL E3; 1.5 mL of solution to each well
4. *A. dahurica* 6.0 µg/mL	3.6 µL *A. dahurica* + 116.4 µL DMSO + 11,880 µL E3; 1.5 mL of solution to each well
5. *A. dahurica* 9.0 µg/mL	5.4 µL *A. dahurica* + 114.6 µL DMSO + 11,880 µL E3; 1.5 mL of solution to each well
6. *A. pubescens* 1.5 µg/mL	0.9 µL *A. pubescens* + 119.1 µL DMSO + 11,880 µL E3; 1.5 mL of solution to each well
7. *A. pubescens* 6.0 µg/mL	3.6 µL *A. pubescens* + 116.4 µL DMSO + 11,880 µL E3; 1.5 mL of solution to each well
8. *A. pubescens* 9.0 µg/mL	5.4 µL *A. pubescens* + 114.6 µL DMSO + 11,880 µL E3; 1.5 mL of solution to each well

**Table 5 ijms-26-02884-t005:** Data on the primers used in the experiment.

Symbol of the Gene	Name of the Gene	Sequence 5′ -> 3′		Amplicon Size (bp)	NCBI Reference Sequence	ACCESSION
		Forward	Reverse			
*hcrt*	hypocretin (orexin) neuropeptide precursor	GACGCAGAAACGACTCTTCC	GGCTTGATTCCGTGAGTTGT	152	NM_001077392.2	NM_001077392
*ccka*	cholecystokinin a	CCAGCTCTCTCTGCGTCTCT	GGTTTGGTCAGCAGGTTGAT	217	NM_001386383.1	NM_001386383 XM_001346104
*crhr1*	Corticotropin-releasing hormone receptor 1	CATAATTCGCCCTGCTGATT	GATGGAGGATGCGACTCATT	197	XM_691254.6	XM_691254
*ist1*	IST1 factor associated with ESCRT-III	ACCTGAACACCAAAGGTTGC	GGAGCAGTGAAAGAGCAAGG	237	NM_212585.2	NM_212585 XM_697451
*avp*	arginine vasopressin	AGAGAGCTGCGCTGTAGACC	TTACAGTGATGTGGGGGACA	157	NM_178293.2	NM_178293
*nr3c1*	nuclear receptor subfamily 3, group C, member 1 (glucocorticoid receptor)	TTCTACGTTGCTGACGATGC	CCGGTGTTCTCCTGTTTGAT	239	NM_001020711.3	NM_001020711 XM_005173120 XM_696817
*nr3c2*	nuclear receptor subfamily 3, group C, member 2	ATTGGGCCTAGTGCAAAATG	TCTCTGTTTGGCTCGGTCTT	249	NM_001100403.1	NM_001100403 XM_685568
*eef1a1l1*	eukaryotic translation elongation factor 1 alpha 1, like 1	GATGCACCACGAGTCTCTGA	TGACCTGAGCGTTGAAGTTG	158	NM_131263.1	NM_131263 XM_001331218
*rpl8*	ribosomal protein L8	GGAGCTCCTCTGGCTAAGGT	CAGGCTTCTCCTCCAGACAG	199	NM_200713.1	NM_200713
*actb1*	actin, beta 1	CTCTTCCAGCCTTCCTTCCT	CTTCTGCATACGGTCAGCAA	165	NM_131031.2	NM_131031

**Table 6 ijms-26-02884-t006:** The concentration (µg RNA/mL) and A260/A280 ratio of samples.

Sample	Concentration (µg RNA/mL)	A260/A280
DMSO control	788.4	1.875
Diazepam control	662.4	1.976
*A. dahurica* 1.5 µg/mL	718.4	2.000
*A. dahurica* 6 µg/mL	695.2	1.946
*A. dahurica* 9 µg/mL	842.4	1.892
*A. pubescens* 1.5 µg/mL	551.2	1.935
*A. pubescens* 6 µg/mL	629.6	1.931
*A. pubescens* 9 µg/mL	569.2	1.955

## Data Availability

Data will be made available on request.

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
