# Peer review of "Exploring the Toxicity and Therapeutic Potential of A. dahurica and A. pubescens in Zebrafish Larvae: Insights into Anxiety Treatment Mechanisms"

_ijms, 2025, doi:10.3390/ijms26072884_

Round 1

Reviewer 1 Report

Comments and Suggestions for Authors

Dear authors, I have gone through the manuscript entitled "Exploring the Toxicity and Therapeutic Potential of Angelica dahurica and Angelica pubescens in Zebrafish Larvae: Insights into Anxiety Treatment Mechanisms" and it is in the scope of the Journal. This manuscript has an interesting topic given that anxiety is widespread today. However, through the detailed analysis of this manuscript, I noticed some omissions. My comments are listed below:

- Highlight the novelty of the study

- Why wasn't in vitro toxicity done (on cell lines)?

- On what basis were the concentrations of extracts tested selected?

- hpf?

- Line 617: Correct the tableX

- whether the role of selected genes has been explained?

- Line 627: Which version of GraphPad?

- Is hearth bedt mentioned in Methods?

- Indicate in the methods which parts of the plant were used, whether the plant material was purchased or harvested and dried. Provide data according to origin.

- why were concentrations of 1.5, 3 and 9 µg/mL used for some extracts and for others 1.5, 6 and 9 µg/mL?

- Merge all gene expression images into one and label it as Figure 4 (A,B,C...)

- The chemical composition obtained should also be discussed.

- to shorten the conclusion since there is a section Future directions

Author Response

Response to Reviewer 1 Comments

Comments and Suggestions for Authors

Dear authors, I have gone through the manuscript entitled "Exploring the Toxicity and Therapeutic Potential of Angelica dahurica and Angelica pubescens in Zebrafish Larvae: Insights into Anxiety Treatment Mechanisms" and it is in the scope of the Journal. This manuscript has an interesting topic given that anxiety is widespread today. However, through the detailed analysis of this manuscript, I noticed some omissions. My comments are listed below:

- Highlight the novelty of the study

Information highlighting the innovative nature of the study was added to the manuscript at the suggestion of the Reviewer.

- Why wasn't in vitro toxicity done (on cell lines)?

Toxicity studies in our experiment were conducted using the zebrafish larvae model because this organism model allows for a more comprehensive picture of the action of a substance at the level of the whole organism, which is important for assessing potential therapeutic and toxic effects. In contrast to in vitro studies on cell lines, the zebrafish model allows for the simultaneous assessment of systemic toxicity, effects on various organs and systems, such as the cardiovascular, nervous or endocrine systems, and monitoring of dynamic developmental processes, such as substance absorption, metabolism and excretion. A key element of our study was the analysis of the expression of genes related to stress regulation and the HPA axis, which is difficult to perform in in vitro systems.

Additionally, zebrafish allow for rapid and efficient studies under conditions that reflect more complex biological interactions in the organism. Although cell lines can be useful for complementary studies, their limitations in reflecting processes occurring in the whole organism make them a less than optimal choice for our study. In the future, in vitro studies may be used to investigate in more detail the mechanisms of action of substances at the cellular level, but in this project the priority was to use a model that allows for the assessment of systemic effects and multi-system interactions.

- On what basis were the concentrations of extracts tested selected?

The concentrations of the extracts tested were selected based on our preliminary experiments and literature review of safety and bioactivity thresholds for similar plant-derived compounds. The goal was to determine a range of concentrations that would allow for the assessment of both subtoxic and potentially therapeutic effects, as well as the dose-dependent toxicity profile of each extract.

Lower concentrations (e.g., 15 and 30 µg/ml) were selected to represent subtoxic levels that may be relevant for therapeutic applications, while higher concentrations (e.g., up to 200 µg/ml) were included to identify any acute toxic effects or lethal thresholds. This approach provided a comprehensive assessment of the safety and efficacy of the extracts within a range that was biologically relevant to the zebrafish model.

In addition, the concentrations were based on the known bioavailability and solubility of the extracts in aqueous media to reflect realistic exposure scenarios in zebrafish larvae. Thanks to this methodology, it was possible to conduct a reliable evaluation of the potential of extracts as therapeutic agents, taking into account their safety and toxicity at different doses.

- hpf?

hpf - hours post fertilization

- Line 617: Correct the tableX

We would like to thank the reviewer for his attention, this has been corrected.

- whether the role of selected genes has been explained?

The role of selected genes was partially elucidated in this study, focusing on their importance in stress response and regulation of the hypothalamic-pituitary-adrenal (HPA) axis. However, while the observed changes in gene expression provide insight into potential mechanisms of action, further studies are needed to fully elucidate the functional roles of these genes in the context of plant extract exposure. This would include investigating their downstream pathways, interactions with other genes, and specific contributions to the observed phenotypic effects, such as cardiotoxicity or anxiolytic properties. We believe that further studies are needed to elucidate the causal mechanisms.

- Line 627: Which version of GraphPad?

GraphPad Prism 10

- Is hearth bedt mentioned in Methods?

Thank you for your suggestion. The heart rate measurement procedure was described in the Methods section, where it is mentioned that after 96 hpf, the heart rate of zebrafish larvae was assessed. To clarify, we have now expanded the description of this methodology to include further details on the procedure used to measure heart rate, including the use of an automated video tracking system and the conditions under which measurements were taken.

- Indicate in the methods which parts of the plant were used, whether the plant material was purchased or harvested and dried. Provide data according to origin.

Thank You for this important question. The two species used in the studies were purchased from a well-known website: planetherbs.pl. In addition, both roots, as well as Radix Angelicae dahuricae (Bai Zhi) as Radix Angelicae pubescentis (Du Huo), were identified by a botanical specialist.

- why were concentrations of 1.5, 3 and 9 µg/mL used for some extracts and for others 1.5, 6 and 9 µg/mL?

The choice of concentrations, such as 1.5, 3, and 9 µg/ml for some extracts and 1.5, 6, and 9 µg/ml for others, reflects differences in toxicity, bioactivity, and solubility of the extracts. Our preliminary data and the literature on the potency of each extract were the basis for this selection. We wanted to cover a range of doses from nontoxic to potentially toxic. The different sets of concentrations were chosen to better understand the dose-response relationships for each extract, allowing for accurate comparisons and optimal statistical analysis. This approach provides a comprehensive assessment of the effects of each extract.

- Merge all gene expression images into one and label it as Figure 4 (A,B,C...)

As per the Reviewer's suggestion, the figures have been combined.

- The chemical composition obtained should also be discussed.

The chemical characterisation is characteristic for representatives of the Apiaceae family and the Angelica genus: the two species studied contain in the root as main secondary metabolites different types of natural coumarin compounds.

- to shorten the conclusion since there is a section Future directions

At the Reviewer's suggestion, the conclusions have been shortened.

Reviewer 2 Report

Comments and Suggestions for Authors

The manuscript lacks a comprehensive discussion of the protein level and fails to provide an analysis of the interactions between proteins. These deficiencies do not align with the standards of this journal; consequently, the manuscript has been rejected."

Comments on the Quality of English Language

The English could be improved to more clearly express the research.

Author Response

Response to Reviewer 2 Comments

Comments and Suggestions for Authors

The manuscript lacks a comprehensive discussion of the protein level and fails to provide an analysis of the interactions between proteins. These deficiencies do not align with the standards of this journal; consequently, the manuscript has been rejected."

Thank you for your valuable feedback. We understand the importance of providing a comprehensive discussion of protein levels and analyzing protein interactions in the context of our study. Unfortunately, due to the scope and limitations of this investigation, we focused primarily on gene expression changes and their relationship to stress and HPA axis regulation. However, we acknowledge that a more thorough exploration of protein expression and interactions would significantly enhance the manuscript's contribution to the field. We plan to address these aspects in future studies, where we will employ additional methodologies, such as proteomics or immunohistochemistry, to investigate protein levels and their interactions. We hope that these future improvements will align better with the standards of the journal and provide a more holistic understanding of the therapeutic potential of Angelica dahurica and Angelica pubescens.

Reviewer 3 Report

Comments and Suggestions for Authors

This manuscript mainly explores the toxicity and therapeutic potential of Angelica dahurica and Angelica pubescens plant extracts in zebrafish embryo/larval models, especially their potential application in anxiety treatment. The study evaluated the safety of these plants and their regulatory effects on anxiety-related genes through toxicity testing, behavioral studies, and gene expression analysis of the two plant extracts. The study found that Angelica dahurica showed lower toxicity at lower concentrations and had certain anti-stress and anxiolytic effects, while Angelica pubescens showed higher toxicity, especially at higher concentrations, which showed increased mortality and severe deformities. Therefore, Angelica dahurica showed strong therapeutic potential, while Angelica pubescens required more careful dosage management.

Overall, the study design of this article is reasonable, an appropriate model (zebrafish) was used to evaluate the toxicity and therapeutic effects of plant extracts, and the description of the methods section is relatively clear. However, there are some expression problems in the article, and some content needs to be further clarified and simplified, especially the discussion of gene expression data. In the Results and Discussion, although rich data support is provided, the interpretation of some experiments and the comparison with other studies could be more in-depth, especially the dose-dependent effects of plant extracts and the comparison with known drugs such as diazepam. Here are my minor concerns:

  1. Ensure consistent use of terminology. For example, in the methods section, “Angelica dahurica” and “Angelica pubescens” can be consistently referred to by their full name or abbreviation (e.g., “ dahurica” and “A. pubescens”).
  2. In the abstract, when mentioning "LD50", you can briefly explain its meaning (for example: 50% lethal dose), because this may not be clear to people who are not familiar with it.
  3. Similarly, abbreviations like "HPA" can be written out in full the first time they appear to make it easier for unfamiliar readers to understand.
  4. The extraction method section could be further detailed to ensure that others can replicate the results, for example, by specifying the time each plant was extracted and the type of solvent used.
  5. When describing the “fish embryo toxicity test”, you can explain why these concentration ranges were chosen. This will help readers understand the rationale of the experimental design.
  6. The Results section is clearly presented, but the Discussion section could be slightly expanded to include more explanation of the gene expression results and comparison with previous studies.
  7. A more systematic comparison of plant extracts with known anxiolytic drugs, such as diazepam, could better highlight the significance of the experimental results.

Author Response

Response to Reviewer  3 Comments

Comments and Suggestions for Authors

This manuscript mainly explores the toxicity and therapeutic potential of Angelica dahurica and Angelica pubescens plant extracts in zebrafish embryo/larval models, especially their potential application in anxiety treatment. The study evaluated the safety of these plants and their regulatory effects on anxiety-related genes through toxicity testing, behavioral studies, and gene expression analysis of the two plant extracts. The study found that Angelica dahurica showed lower toxicity at lower concentrations and had certain anti-stress and anxiolytic effects, while Angelica pubescens showed higher toxicity, especially at higher concentrations, which showed increased mortality and severe deformities. Therefore, Angelica dahurica showed strong therapeutic potential, while Angelica pubescens required more careful dosage management.

Overall, the study design of this article is reasonable, an appropriate model (zebrafish) was used to evaluate the toxicity and therapeutic effects of plant extracts, and the description of the methods section is relatively clear. However, there are some expression problems in the article, and some content needs to be further clarified and simplified, especially the discussion of gene expression data. In the Results and Discussion, although rich data support is provided, the interpretation of some experiments and the comparison with other studies could be more in-depth, especially the dose-dependent effects of plant extracts and the comparison with known drugs such as diazepam. Here are my minor concerns:

  1. Ensure consistent use of terminology. For example, in the methods section, “Angelica dahurica” and “Angelica pubescens” can be consistently referred to by their full name or abbreviation (e.g., “ dahurica” and “A. pubescens”).

Abbreviations were used as per the Reviewer's suggestion.

  1. In the abstract, when mentioning "LD50", you can briefly explain its meaning (for example: 50% lethal dose), because this may not be clear to people who are not familiar with it.

Following the reviewer's suggestion, this has been clarified.

  1. Similarly, abbreviations like "HPA" can be written out in full the first time they appear to make it easier for unfamiliar readers to understand.

This abbreviation was explained the first time it was used (last paragraph of the introduction).

  1. The extraction method section could be further detailed to ensure that others can replicate the results, for example, by specifying the time each plant was extracted and the type of solvent used.

Thank you for that question. The concerning extraction method all detail were described. The raw materials were then transferred to a 10ml cell, and extracted with methanol by pressurized liquid extraction (ASE). One cycle lasted 10 minutes and each portion of the plant material was extracted three times with a new portion of a given solvent (temp. 80⁰C, pressure up to 110 atm). Each extract obtained was poured into a 50 ml volumetric flask, and made up to that amount with the solvent used for extraction. The extracts were stored in a refrigerator.

  1. When describing the “fish embryo toxicity test”, you can explain why these concentration ranges were chosen. This will help readers understand the rationale of the experimental design.

Thank you for your suggestion. The concentration ranges used in the fish embryo toxicity test were selected based on previous studies and preliminary observations. Our goal was to cover a range from subtoxic to potentially harmful levels to assess both therapeutic effects and toxicity. Lower concentrations were selected to assess potential benefits, while higher concentrations tested toxicity thresholds. This approach allowed us to observe both positive and negative effects on developing zebrafish embryos. The selected ranges were also confirmed by similar studies in the literature and our knowledge of the properties of plant extracts. We have added this fragment to the methods section.

  1. The Results section is clearly presented, but the Discussion section could be slightly expanded to include more explanation of the gene expression results and comparison with previous studies.

Thank you for your suggestion. We have expanded the Discussion section to provide a more detailed explanation of the gene expression results and to include a more thorough comparison with previous studies.

  1. A more systematic comparison of plant extracts with known anxiolytic drugs, such as diazepam, could better highlight the significance of the experimental results.

Thank you for your suggestion. A more systematic comparison of plant extracts with known anxiolytic drugs such as diazepam would indeed provide valuable context for the interpretation of experimental results. In our study, diazepam was included as the reference anxiolytic drug to assess the effects of plant extracts on stress- and anxiety-related gene expression. A more detailed comparison, taking into account factors such as dose-response relationships, time-dependent effects, and molecular mechanisms, would help to highlight the potential therapeutic value of Angelica dahurica and Angelica pubescens. We will build on this suggestion in future studies. In the present work, a comparison of the efficacy of extracts to diazepam is included.